# Orbitally dominated Rashba-Edelstein effect in noncentrosymmetric antiferromagnets

Leandro Salemi[1]*, Marco Berritta [1], Ashis K. Nandy[1,2] & Peter M. Oppeneer [1]*

Efficient manipulation of magnetic order with electric current pulses is desirable for achieving fast spintronic devices. The Rashba-Edelstein effect, wherein spin polarization is electrically induced in noncentrosymmetric systems, provides a mean to achieve staggered spin-orbit torques. Initially predicted for spin, its orbital counterpart has been disregarded up to now. Here we report a generalized Rashba-Edelstein effect, which generates not only spin polarization but also orbital polarization, which we find to be far from being negligible. We show that the orbital Rashba-Edelstein effect does not require spin-orbit coupling to exist. We present first-principles calculations of the frequency-dependent spin and orbital Rashba-Edelstein tensors for the noncentrosymmetric antiferromagnets CuMnAs and $Mn_2Au$. We show that the electrically induced local magnetization can exhibit Rashba-like or Dresselhaus-like symmetries, depending on the magnetic configuration. We compute sizable induced magnetizations at optical frequencies, which suggest that electric-field driven switching could be achieved at much higher frequencies.

[1] Department of Physics and Astronomy, Uppsala University, P. O. Box 516, S-751 20 Uppsala, Sweden. [2]Present address: School of Physical Sciences, National Institute of Science Education and Research, HBNI, Jatni 752050 Odisha, India. *email: leandro.salemi@physics.uu.se; peter.oppeneer@physics.uu.se

The efficient manipulation of the magnetization of materials remains a crucial challenge in the field of spintronics[1–3]. Although previously disregarded, antiferromagnets have recently emerged as appealing candidate materials for information storage devices since they offer various advantages[4–6]. Specifically, antiferromagnets are robust against external magnetic field perturbations, they are available as insulators, semiconductors, and metals, allowing thus for versatile environment integration, and they often have a high Néel temperature[7–9], suitable for room-temperature operation of the devices. Moreover, their intrinsic spin dynamics is ultrafast, in the THz domain[10–12] (compared with GHz dynamics reported for ferromagnets[13,14]).

Achieving efficient control over the magnetization in antiferromagnets is however an entirely different issue. The spin Hall effect has proven to generate a spin-polarized current[15,16] and thereby create a spin–orbit torque that can act efficiently on the magnetization of a ferromagnetic layer[17–20]. A different effect, the Rashba-Edelstein effect (REE) has been proposed as a method to induce a nonequilibrium spin polarization through an electrical current in solids lacking inversion symmetry[21]. This effect (also called inverse spin galvanic effect) was initially predicted in 1990 by Edelstein[21] using a Rashba spin–orbit coupling (SOC)[22]; it was experimentally observed in GaAs heterostructures[23–25].

More recently, the REE has been proposed as a method to create a current-induced staggered spin polarization and spin–orbit torque in the noncentrosymmetric antiferromagnets CuMnAs and Mn$_2$Au (refs. [9,26,27]) causing the antiferromagnetic magnetic moments to flip to a perpendicular direction[28–33]. These recent experiments have shown that current-driven switching of the Néel vector is possible, however, the operation of the spin–orbit torque and the switching path are not understood yet. Microscopic investigations indicate that a complex switching process with both domain wall motion and domain flips may occur[34,35]. It is moreover a question how large the induced staggered moments are. So far, linear-response tight-binding calculations with Rashba SOC[36,37] and an ab initio calculation[28] of the current-induced magnetic fields have been performed, that however differed considerably. Also a semiclassical model based on the Boltzmann equation has been employed to compute the induced magnetization in a Weyl semimetal[38]. These investigations concentrate moreover on the induced spin polarization and neglect any possible contribution stemming from an induced orbital magnetization.

Here, we investigate computationally the full magnetic polarization induced by a time-varying applied electric field in the noncentrosymmetric antiferromagnets CuMnAs and Mn$_2$Au. Our density functional theory (DFT)-based calculations bring insights into the REE in these antiferromagnets. We show that the dominant contribution to the induced polarizations stems from the orbital Rashba-Edelstein effect (OREE). The OREE tensor can have a symmetry different from that of the spin Rashba-Edelstein effect (SREE) tensor (e.g., Rashba vs. Dresselhaus-type of symmetry). Due to the pronounced Rashba symmetry of the OREE tensor, a strong orthogonal orbital-momentum locking is obtained for in-plane electric fields. We find furthermore that quite sizable moments can be electrically induced on the nonmagnetic atoms. Investigating the origin of the large induced orbital polarizations, we show that these are present even without spin–orbit interaction, whereas the spin REE tensor is proportional to the SOC and vanishes without SOC, signifying that the latter is induced through the relativistic SOC, whereas the former have a non-relativistic origin.

## Results

**Theoretical framework.** We use linear-response theory to evaluate the magnetic response to a time-dependent electric field $\mathbf{E}(t)$. The induced magnetic polarization $\delta\mathbf{M} = \mu_B\delta(\mathbf{L} + 2\mathbf{S})$, consisting of orbital ($\mathbf{L}$) and spin ($\mathbf{S}$) contributions, reads in the frequency domain

$$\delta M_i(\omega) = \sum_j \left[\chi_{ij}^L(\omega) + 2\chi_{ij}^S(\omega)\right]E_j(\omega), \quad (1)$$

where $\chi_{ij}^L(\omega)$ and $\chi_{ij}^S(\omega)$ ($i$, $j$ = x, y, or z) are the orbital and spin Rashba-Edelstein susceptibility tensor, respectively (in units of $\mu_B$ nm V$^{-1}$). For the response-theory expressions for the OREE and SREE tensors, see Methods subsection linear-response-theory formulation.

To evaluate the frequency-dependent SREE and OREE tensors, we adopt the DFT formalism as implemented in the full-potential linearized augmented plane-wave (FLAPW) all-electron code WIEN2k[39]. For details of the computational approach, see the Methods subsection details on the numerical calculations. In the following section, we apply this framework to noncentrosymmetric CuMnAs and Mn$_2$Au that have recently drawn attention for antiferromagnetic spintronics[4,6,28,30,31,33].

**Results for CuMnAs.** Our DFT calculations give that CuMnAs has an antiferromagnetic (AFM) ground state with Mn atoms carrying a magnetic moment of ∼3.66 $\mu_B$, in agreement with recent experiments[26,27]. The tetragonal cell of CuMnAs (space group $P4/nmm$), shown in Fig. 1a, consists of six inequivalent atoms, two of each chemical species. Both the Mn and As atoms have the $4mm$ point group, whereas the Cu atoms possess the $-4m2$ point group symmetry. The magnetic ordering is such that adjacent {001} Mn planes are antiferromagnetically coupled while Mn atoms laying in the same plane are ferromagnetically ordered. The As atoms are also found to carry a small magnetic moment of ∼2.33 10$^{-3}$ $\mu_B$. Their orientation is such that {001} As planes are ferromagnetically coupled to the closest {001} Mn plane. The Cu atoms are found to be nonmagnetic.

The AFM moments can orient along different Néel vector axes, and this direction of the AFM moments depends sensitively, for thin films, on the interplay of intrinsic magneto-crystalline anisotropy and shape anisotropy. Experimentally, an orientation of the spins in the $ab$ plane has been observed for thin films[27].

The REE tensors depend on the orientation of the moments. We compute them here for different orientations of the moments, and in addition, we compute the atom-resolved tensors' spectra, using a specific labeling of the atoms as shown in Fig. 1a.

We first consider the case where magnetic moments are oriented along the $c$-axis and the applied field is along the $a$-axis (Fig. 1a). This magnetic configuration does not break the fourfold rotational symmetry about the $c$-axis (hard magnetization axis). The real parts of the nonzero components of the atom-resolved spin and orbital Rashba-Edelstein tensors are displayed in Fig. 1d. The imaginary parts of the Rashba-Edelstein tensors are given in Supplementary Fig. 1. Several remarkable observations can now be made. First, there are frequency-dependent induced moments not only on the Mn atoms but also on the Cu and As atoms. Second, the orbital contribution, that was thus far disregarded, is not negligible. In fact the staggered orbital part $\chi_{xy}^L$ is the dominating part of the response and is ∼45 times larger than its spin counterpart at $\omega = 0$. In the near-infrared region ($\hbar\omega = 0.6$–1.2 eV) $\chi_{xy}^L$ dominates even more, since the spin response $\chi_{xy}^S$ is quite small. Third, apart from the staggered components, that are such that antiferromagnetic Mn$_1$ and Mn$_2$

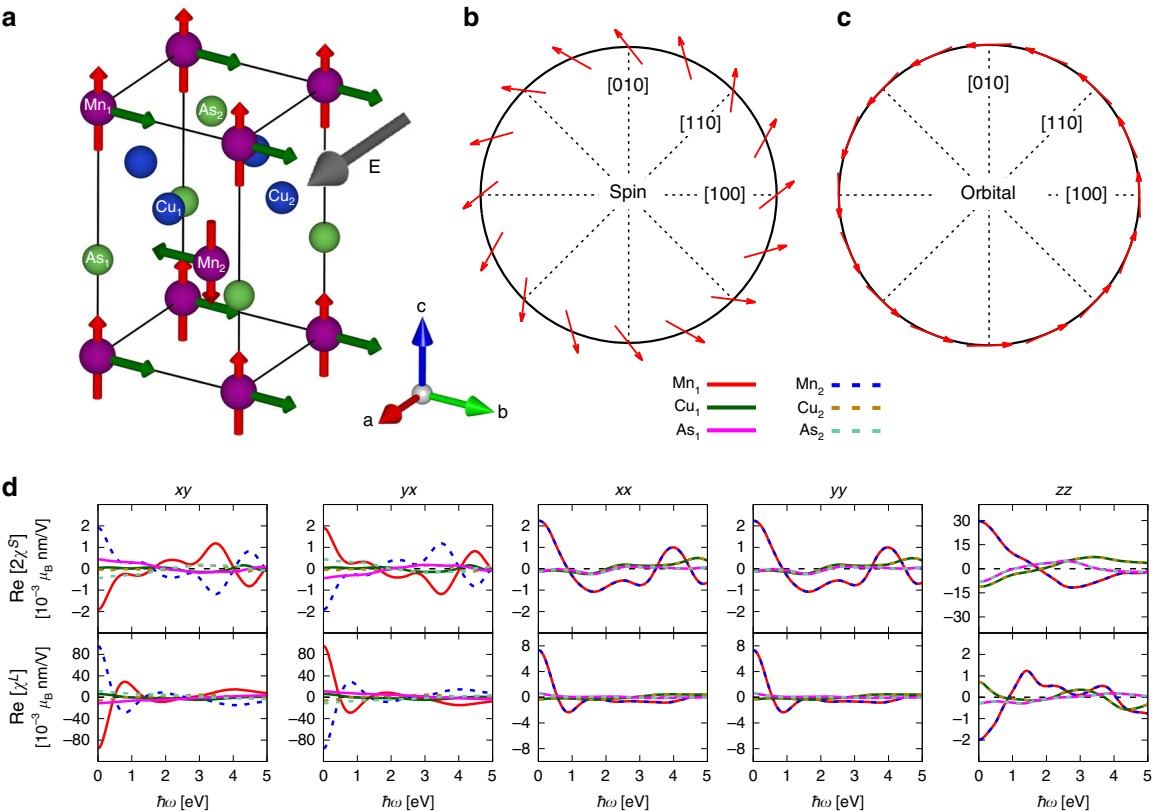

**Fig. 1** Magnetization induced by the Rashba-Edelstein effect in antiferromagnetic CuMnAs. **a** Sketch of the tetragonal unit cell of CuMnAs with the magnetic moments constrained to the $c$-axis. The inset depicts the direction of the crystal axes. The red arrows on the manganese atoms represent the initial magnetic moments. Applying an electric field **E** along the (100) direction (gray arrow) induces a nonequilibrium magnetization mainly along the (010) direction (green arrows). **b** Symmetry of the induced spin magnetization as a function of the static electric-field direction for Mn$_1$. **c** Symmetry of the induced orbital magnetization as a function of the static electric-field direction for Mn$_1$. **d** Real parts of the nonzero tensor components (labeled $ij$, with $i$, $j = x$, $y$, or $z$) of the spin and orbital Rashba-Edelstein susceptibilities, Re$[2\chi^S]$ and Re$[\chi^L]$, as function of the electric-field frequency, $\hbar\omega$.

atoms experience an opposite response (off-diagonal xy and yx), there are also homogeneous induced components that act in the same direction for a given atomic species (see diagonal xx, yy, and zz tensors elements in Fig. 1d). These non-staggered induced longitudinal magnetizations are not small, especially for the spin response, and can alter the atomic torques and influence eventual spin switching. Lastly, we note that SREE and OREE tensors of the individual elements obey different symmetries, specific to the atomic site's point group. For the Mn atoms, we observe $\chi_{xy}^{S,L} = -\chi_{yx}^{S,L}$, and $\chi_{xx}^{S,L} = \chi_{yy}^{S,L}$. The same symmetry of the tensors is obtained for the As atoms, but for the Cu atoms $\chi_{xy}^{S,L} = \chi_{yx}^{S,L}$ and $\chi_{xx}^{S,L} = \chi_{yy}^{S,L}$.

The calculated orientation of the induced moments as a function of the direction of an in-plane applied static electric field is displayed in Figs. 1b, c for the spin and orbital part, respectively. We observe a Rashba-like behavior for the spin part with nonorthogonal spin-momentum locking, whereas the orbital part possesses a nearly perfect Rashba symmetry with orthogonal orbital-momentum locking (for a definition, see refs. [40,41]). These plots are obtained by computing the tensors at $\omega = 0$, while varying the direction of **E**. It is important to note that the induced spin and orbital moments depend on the frequency $\omega$. In addition, the fact that the spin and orbital polarization are induced in different directions, and can even be antiparallel (see below) has an important consequence. The resultant torque field that acts on the atomic moments in a Landau–Lifshitz–Gilbert spin-dynamics formulation can then not be represented in the form of a single-atomic Zeeman field, corresponding to an

interaction $(\mu_B/\hbar)(\hat{\mathbf{L}} + 2\hat{\mathbf{S}}) \cdot \mathbf{H}$, with **H** the applied atomic Zeeman magnetic field, as this would lead to a proportional induced spin and orbital atomic moment.

We now consider the case of CuMnAs with an in-plane magnetization along the (100) direction which corresponds to the magnetic structure realized in experiments. As shown in Fig. 2a, applying a static electric field (gray arrow) along the magnetization direction (red arrows) induces magnetic moments (green arrows) mainly on the Mn atoms. Those magnetic moments are staggered, i.e., they are practically antiparallel to each other for AFM coupled Mn atoms. However, a parallel out-of-plane contribution is also present. This non-staggered feature of the magnetic response can be recognized by looking at the SREE and OREE tensors, shown in Fig. 2d. Especially, the nonzero $\chi_{zx}^S$ tensor component gives a non-staggered contribution. Nonetheless, the by far dominant part of the induced magnetic polarization is again contained in the staggered xy and yx components of the orbital response. The imaginary parts of the susceptibility tensors are given in Supplementary Fig. 2.

Another important point to be noticed is that the nonzero homogeneous tensor components have changed with the changed direction of the Néel vector. The non-staggered components for CuMnAs with magnetization along (001) were the diagonal xx, yy, and zz components, while for the magnetization along (100) these are the xz and zx components. As can be seen in Figs. 1d and 2d, the computed SREE spectra are very similar, with an inverted sign ($\chi_{zz}^S \longrightarrow \chi_{xz}^S$, and $\chi_{xx}^S \longrightarrow -\chi_{zx}^S$). This is a direct demonstration that the electrically induced magnetization texture

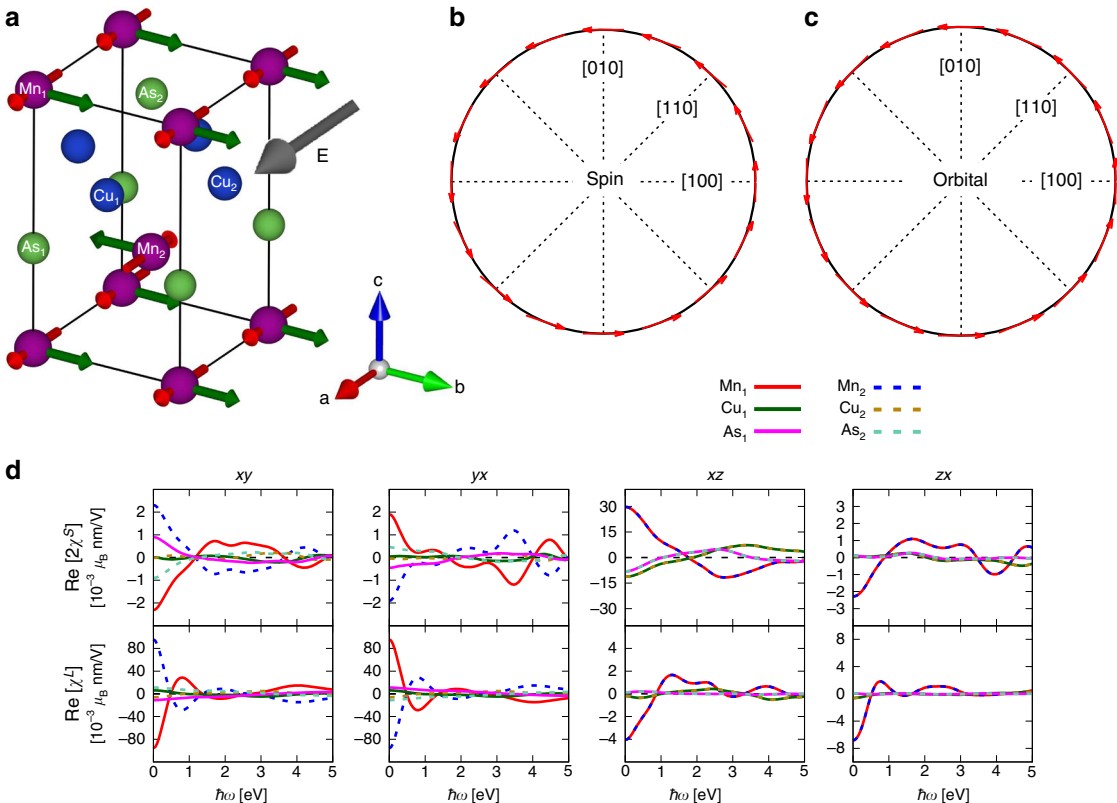

**Fig. 2** Rashba-Edelstein effect in CuMnAs with magnetic moments along the *a*-axis. **a** Sketch of the tetragonal unit cell of CuMnAs. The red arrows on the Mn atoms represent the initial magnetic moments. Applying an electric field **E** along the (100) direction (gray arrow) induces a nonequilibrium magnetization mainly along the (010) direction (green arrows). **b** In-plane symmetry of the induced spin magnetization as a function of the static electric-field direction for $Mn_1$. **c** In-plane symmetry of the induced orbital magnetization as a function of the static electric-field direction for $Mn_1$. **d** Real parts of the nonzero components *ij* (*i*, *j* = *x*, *y*, or *z*) of the spin and orbital Rashba-Edelstein susceptibility tensors, $Re[2\chi^S]$ and $Re[\chi^L]$, as function of the driving electric-field frequency, $\hbar\omega$.

depends on the underlying magnetization direction itself. This can be understood as an influence of the magnetization direction on the eigenstates, which affects the induced magnetization[37]. This effect has also been observed experimentally in (Ga, Mn) As[42]. Computing the symmetry of the momentum-dependent induced spin and orbital polarizations for an in-plane static electric field, we find that both the spin- and orbital-resolved parts exhibit a Rashba symmetry (Fig. 2b, c). Here, it can be recognized that the induced spin and orbital polarizations cooperate and exert spin and orbital torques in the same direction. We further note that the symmetries of the REE tensor are now such that $\chi^S_{xy} \neq -\chi^S_{yx}$, but $\chi^L_{xy} = -\chi^L_{yx}$ for the Mn and As atoms. The latter tensor elements are the largest, which illustrates the dominance of the orbital REE.

**Results for $Mn_2Au$**. $Mn_2Au$ crystallizes in the tetragonal structure shown in Fig. 3a ($I4/mmm$ space group). The ground state of $Mn_2Au$ is computed to be antiferromagnetic with magnetic moments of 3.69 $\mu_B$ only on the manganese atoms. Experimentally, the magnetization of $Mn_2Au$ films is found to lie in {001} (basal) planes, with ~4 $\mu_B$ moments on Mn[9]. The unit cell consists of two equivalent Au atoms and two pairs of inequivalent Mn atoms, labeled $Mn_1$ and $Mn_2$ in Fig. 3a. The four Mn atoms have the $4mm$ (polar) point group symmetry, and the two Au atoms have the $4/mmm$ (centrosymmetric) point group symmetry.

Figure 3d shows the real parts of the nonzero SREE and OREE tensor elements, computed for Mn moments along the *c*-axis. The

imaginary parts of the susceptibility tensors are given in Supplementary Fig. 3. The calculated tensors exemplify that the REE of $Mn_2Au$ is in several aspects different from that in CuMnAs. The spin and orbital responses for both the xy and yx components are staggered in $Mn_2Au$, and the homogeneous part of the response is in the diagonal xx, yy, and zz components, similar to CuMnAs. The orbital part of the response is not as dominant as in the case of CuMnAs. The largest orbital contribution in the off-diagonal elements is almost 12 times larger than the spin contribution for $\omega = 0$. We can furthermore observe that the nonmagnetic Au atoms do not display any finite staggered response, consistent with the centrosymmetric nature of its point group $4/mmm$.

The directional dependence of the current-induced moments on Mn atoms as a function of the direction of an in-plane applied static electric field is shown in Fig. 3b, c for the spin and orbital response, respectively. The spin response exhibits a Rashba-like behavior, and the orbital counterpart possesses a Rashba-like symmetry, too, but notably practically in a direction perpendicular to that of the spin response. Hence, for any applied in-plane field, the current-induced spin and orbital moments will exert torques in perpendicular directions during a switching process.

We now consider $Mn_2Au$ with moments laying in the *ab* plane along the (100) direction, see Fig. 4a. As for CuMnAs, the magnetic moments have been experimentally found to lie in the *ab* plane[9]. Here, the calculated nonzero REE tensor elements are the xy, yx, xz, and zx components. The real parts of the nonzero Rashba-Edelstein tensor elements are shown in Fig. 4d, and the imaginary parts are given in Supplementary Fig. 4. In this

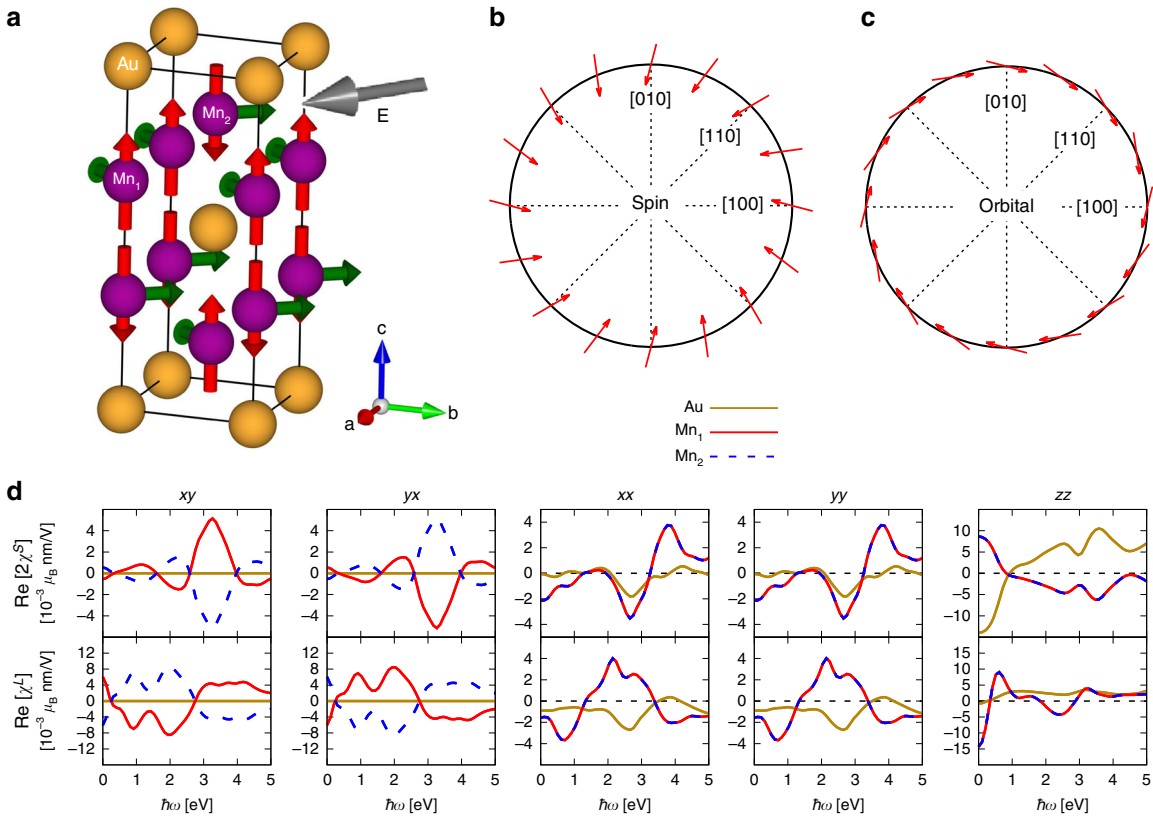

**Fig. 3** Rashba-Edelstein effect in $Mn_2Au$ with magnetic moments along the $c$-axis. **a** The unit cell of $Mn_2Au$, with red arrows on the Mn atoms depicting the initial magnetic moments. Applying an electric field **E** along the (100) direction (gray arrow) induces a nonequilibrium magnetization tilted in between the (010) and (100) direction (green arrows). **b** Symmetry of the induced spin magnetization as a function of the static electric-field direction for $Mn_1$. **c** Symmetry of the induced orbital magnetization as a function of the static electric-field direction for $Mn_1$. **d** Real parts of the nonzero components $ij$ ($i$, $j$ = $x$, $y$, or $z$) of the spin and orbital Rashba-Edelstein susceptibilities, $Re[2\chi^S]$ and $Re[\chi^L]$, as function of electric-field frequency, $\hbar\omega$.

configuration, the staggered responses for both spin and orbital contributions are present in the xy and yx components, whereas the non-staggered responses are present in the zx and xz components, that however give smaller contributions. The mainly staggered response corroborates the investigation of Želzený et al.[36] who predicted staggered spin–orbit fields on the two Mn sublattices. The symmetries of the main staggered tensor elements are $\chi_{xy}^S \neq -\chi_{yx}^S$ but $\chi_{xy}^L = -\chi_{yx}^L$, as we also obtained for CuMnAs with Néel vector along the $a$-axis.

Considering the symmetry of the induced polarizations for an in-plane field in Fig. 4b, c, we find that the SREE exhibits a Dresselhaus-type behavior and the OREE exhibits a pure Rashba symmetry. Again, the possible non-cooperativity of the OREE and SREE when exerting a torque can be fully recognized. When both the static moments and electric field are along the $a$-axis ([100]), the induced spin and orbital magnetizations are antiparallel and the torques will partially compensate each other. For an in-plane electric field **E** along the $b$-axis, the orbital and spin magnetizations do not counteract each other (see also Fig. 4d), but this configuration only leads to an induced longitudinal moment along the static AFM moments that does not exert a local torque on the atomic moment. This exemplifies that devising optimal switching conditions can be quite intricate.

## Discussion

Previously, Edelstein predicted an electrically induced out-of-equilibrium spin magnetization generated by Rashba SOC[21]. Here, without assuming any specific shape for the SOC, we find that, depending on the magnetic configuration, the symmetry of

the induced magnetization can adopt Rashba-like or Dresselhaus-like behaviors. Remarkably, we find that the previously neglected orbital polarization can in fact be much larger than the induced spin polarization. The possible existence of an induced orbital magnetization has been suggested in earlier studies[43,44].

The importance of SOC on the magnetoelectric susceptibilities can be accessed by reducing or switching off SOC in the calculations. The results of these calculations are shown in Fig. 5. We find that the SREE computed without SOC completely vanishes; therefore, consistent with Edelstein 1990[21], this is an intrinsic effect which occurs due to the broken local inversion symmetry in the presence of SOC. Surprisingly, however, for the orbital component our calculations without SOC give an unchanged, non-vanishing OREE response for the dominant off-diagonal tensor elements. In Fig. 5a, b, we show the computed SOC dependence of the xx and xy tensor elements of the SREE susceptibility of $Mn_1$ in CuMnAs with antiferromagnetic moments along the $c$-axis. These elements decrease linearly with decreasing SOC. For the OREE, we find that the staggered components $\chi_{xy}^L$, shown in Fig. 5d, and $\chi_{yx}^L$ (not shown), are present even without SOC, and are not even changed by SOC strength which suggests that the leading off-diagonal term is independent of SOC. In contrast, without SOC the non-staggered OREE components $\chi_{xx}^L$, shown in Fig. 5c, as well as $\chi_{yy}^L$ and $\chi_{zz}^L$ (not shown) vanish, and these can consequently be identified as intrinsic SOC-related quantities. This observation is quite crucial and unexpected, since the staggered SREE components are generally believed to be at the origin of switching, in, e.g., CuMnAs, and to be SOC related. We find that the dominant nonrelativistic contribution is in the

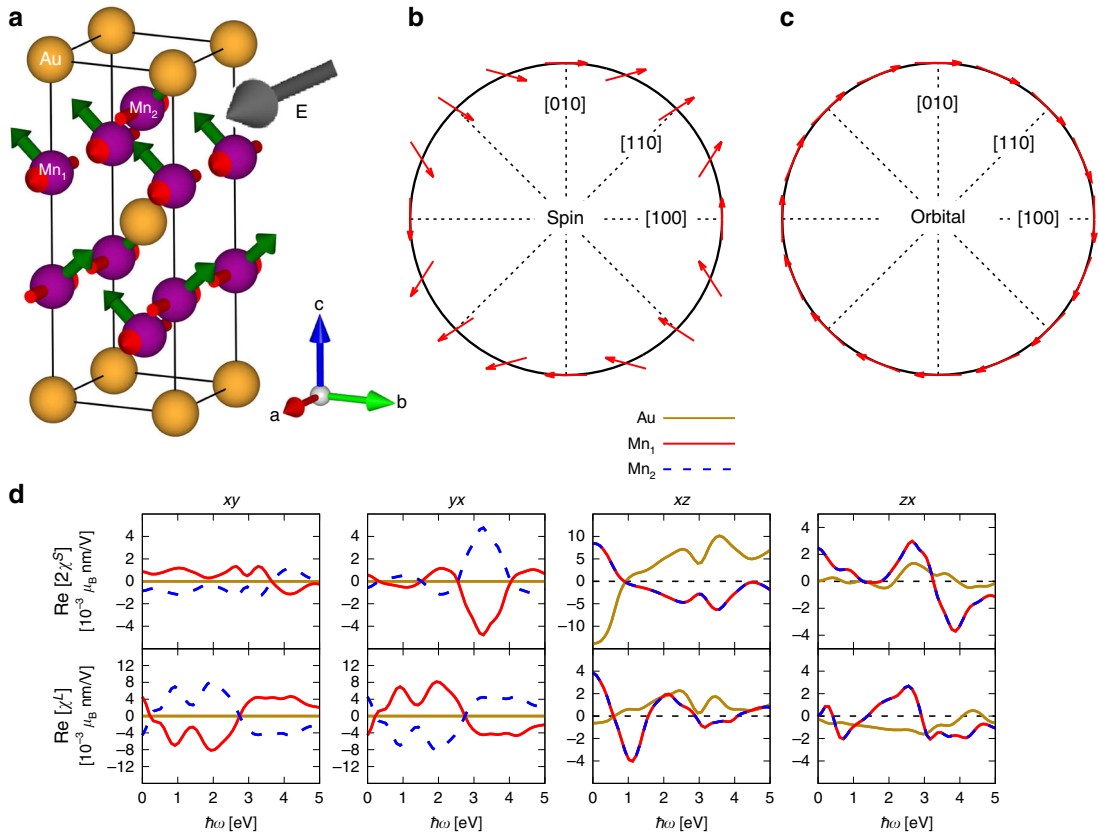

**Fig. 4** Rashba-Edelstein effect in $Mn_2Au$ with magnetic moments along the $a$-axis. **a** The unit cell of $Mn_2Au$. The red arrows on the Mn atoms represent the initial magnetic moments. Applying an electric field **E** along the (100) direction (gray arrow) induces a nonequilibrium magnetization mainly along the (010) direction (green arrows). **b** In-plane symmetry of the induced spin magnetization as a function of the static electric-field direction for $Mn_1$. **c** Symmetry of the induced orbital magnetization as a function of the static electric-field direction for $Mn_1$. **d** Real parts of the nonzero tensor components $ij$ ($i$, $j = x$, $y$, or $z$) of the spin and orbital Rashba-Edelstein susceptibilities, $Re[2\chi^S]$ and $Re[\chi^L]$, as function of the electric-field frequency, $\hbar\omega$.

staggered OREE components while smaller staggered spin components and non-staggered orbital components are generated by SOC.

As yet we know little about the influence of the OREE for a magnetization switching event, but a cautioning remark is warranted. Although the OREE can be large, to act on the spin moments present in an AFM, it needs to couple to these through spin–orbit interaction. Then, the overall torque on the antiferromagnetic spin moments will eventually be proportional to the SOC.

To analyze the origin of the induced orbital polarizations, we observe that due to the staggered nature of the induced moments in Fig. 3, the sum of the induced orbital moments on all atoms in the unit cell cancels, but the contributions on individual atoms do not. There is thus an atomic orbital polarization present even without SOC. The orbital angular momentum dynamics induced by the applied potential $\hat{V}(t) = -e\,\mathbf{E}(t)\cdot\hat{\mathbf{r}}$, where e is the electron charge and $\hat{\mathbf{r}}$ the position operator, can be evaluated from the Heisenberg expression in a single-electron picture as

$$\frac{d\hat{\mathbf{L}}^{ind}}{dt} = -\frac{i}{\hbar}\left[\hat{\mathbf{L}}^{ind}, \hat{V}(t)\right] = \hat{\mathbf{r}}\times e\,\mathbf{E}(t), \qquad (2)$$

which is the quantum mechanical counterpart of the classical equation of motion for angular momentum, $\frac{d\mathbf{L}}{dt} = \mathbf{r}\times\mathbf{F}$, where **F** is an externally applied force. In this picture, the electric field acts as a torque on the center of mass of the electrons on an atom. The current-induced nonequilibrium electron populations provide then a nonzero atomic orbital polarization, similar to the

nonequilibrium Fermi surface shift leading to spin–orbit torques[42]. This mechanism does not require the interplay of SOC, as the field couples directly to the position of the electrons and thereby affects the orbital momentum. Therefore, the OREE does not arise only from the small relativistic SOC, and sizeable effects might thus even be observed in systems with small SOC.

A symmetry analysis adds further insight to the appearance of the orbital texture even without SOC. In the crystal structure of antiferromagnetic tetragonal CuMnAs, inversion symmetry $\mathcal{P}$ and time-reversal symmetry $\mathcal{T}$ are broken, but $\mathcal{PT}$ symmetry is obeyed. The antiferromagnetic Mn atoms $Mn_1$ and $Mn_2$ are inversion partners under $\mathcal{PT}$ symmetry[28]. Using $\mathcal{P}$ and $\mathcal{T}$ transformation properties, $\mathcal{PT}\{L_{Mn_1}^{ind}\} = -L_{Mn_2}^{ind}$ and vice versa, i.e., $\mathcal{PT}$ symmetry enforces the nonzero orbital moments to be staggered. Even with vanishing equilibrium spin moments the two Mn atoms are inversion symmetry partners and, interestingly, $\mathcal{PT}$ symmetry enforces the induced orbital polarizations to be staggered as well in the nonmagnetic phase. An example of the calculated SREE and OREE tensors of nonmagnetic CuMnAs is given in Supplementary Fig. 5.

We furthermore point out that the here-observed appearance of an orbital polarization in the unit cell in the absence of SOC is distinct from other recent theoretical predictions of nonzero orbital textures[45–47]. Hanke et al.[46] showed that a nonzero static orbital moment can arise in the noncoplanar antiferromagnet $\gamma$-FeMn without SOC due to spin chirality. Here, in the absence of spin chirality, we predict nonzero orbital moments that are present without SOC but permitted by $\mathcal{PT}$ symmetry when an applied electric field is present. Yoda et al.[47] proposed that in a

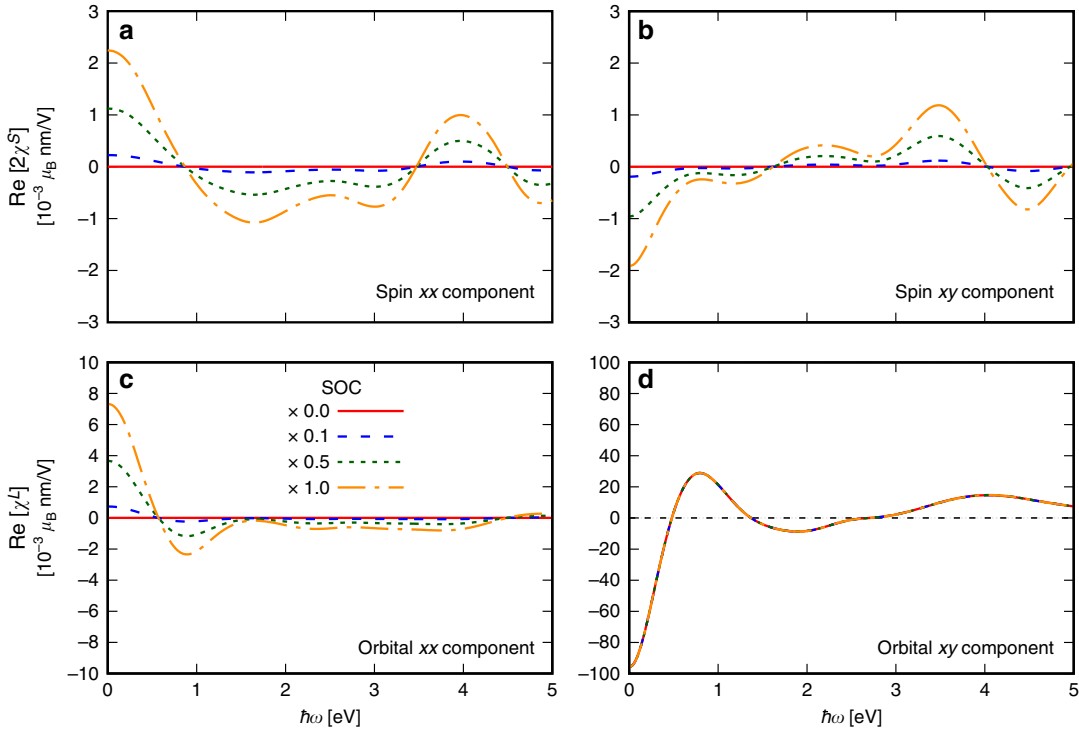

**Fig. 5** Rashba-Edelstein effect as a function of spin–orbit coupling strength. Calculated results are given for the $Mn_1$ atom of antiferromagnetic CuMnAs with moments along the $c$-axis, for scaled values of the SOC, as given in the legend. **a**, **b** The results for the $Re[\chi^S_{xx}]$ and $Re[\chi^S_{xy}]$ component, respectively, of the spin REE susceptibility tensor. **c**, **d** The results for the $Re[\chi^L_{xx}]$ and $Re[\chi^L_{xy}]$ component of the orbital REE tensor, respectively, as function of electric-field frequency $\hbar\omega$.

chiral crystal the solenoidal electron hopping motion could lead to an orbital magnetization in systems with time-reversal symmetry, in the sense of an orbital Edelstein effect. Our here-computed induced orbital polarization is distinctly different, as it does not require chiral crystal symmetry.

To analyze which orbitals contribute most to the Rashba-Edelstein tensors, we perform a projection of the OREE and SREE tensors in Eq. (3) on the atomic-orbitals of the constituting atoms. The computed orbital-projected SREE and OREE susceptibilities are shown in Supplementary Figs. 6 and 7 for CuMnAs and $Mn_2Au$, respectively, with moments along the $c$-axis. These calculations demonstrate that the contributions from $s$ orbitals are negligible for the spin susceptibility tensors and exactly zero for the orbital susceptibility. The contributions from $p$ orbitals are also very small. The full response stems nearly completely from the $d$ orbitals. Scanning the Fermi filling energy through the energy window of the Mn $d$ orbitals from $-4$ eV below the Fermi energy to 4 eV above the Fermi energy reveals furthermore that the main staggered orbital tensor elements tend to follow the size of the Mn spin-polarized density of states (DOS). This implies that the SREE and OREE will be largest when the Fermi energy is close to a maximum of the $3d$ DOS, a feature that can offer pathways to fabricate materials that display large induced spin and orbital polarizations.

Bandfilling modifies the sizes of both the SREE and OREE tensor elements and therefore also modifies the symmetry character of the induced spin and orbital textures. This happens in a materials' specific way, yet it is possible to obtain bandfillings where both the induced spin and orbital polarizations exhibit a nearly perfect Rashba texture and are moreover parallel to one another (cf. Fig. 2). Other bandfillings can change a Rashba-type texture into a Dresselhaus texture. An example of such situation is given in Supplementary Fig. 8, where the SREE and OREE tensors are computed for CuMnAs with moments along the $a$-axis for a shift of the chemical potential of 0.6 eV; at this band-filling the SREE tensor exhibits a Dresselhaus-like symmetry and the OREE tensor a Rashba symmetry.

Our calculations predict sizable induced polarizations at finite frequencies, which raises the question whether electric-field driven magnetic moment switching could be achieved at high frequencies. It is well known that time-dependent magnetic fields cannot drive fast spin dynamics of ferromagnets in the optical regime because the magnetic permeability $\mu(\omega)$ decays quickly as $\omega$ increases to the infrared region[48]. The situation is however entirely different for the SREE and OREE. The magnetic permeability is due to a magnetic field $\mathbf{H}$ that acts on the spin through the Zeeman interaction in the Hamiltonian, $\mu_B \hat{\boldsymbol{\sigma}} \cdot \mathbf{H}(t)$, whereas for the SREE the electric field couples to the charge, $-e \hat{\mathbf{r}} \cdot \mathbf{E}(t)$. The electric charges can indeed follow the rapidly changing $E$-field, implying that an equally fast magnetic response can be anticipated. Due to their electrical origin, the induced magnetizations can be driven at petahertz frequencies, thus opening for potential routes to achieve petahertz spintronics. This would be so particularly for thin functional devices, since the penetration depth of optical fields is typically $\leq 20$ nm.

In the DC limit, $\omega = 0$, the real part of the REE is nonzero and its imaginary part vanishes exactly. At finite frequencies, both the real and imaginary parts of the tensor components can be nonzero. The nonzero imaginary REE susceptibility has a specific influence on the evolving magnetization dynamics. For a given driving electric field $\mathbf{E}(t)$, the induced spin polarization $\delta \mathbf{M}^S(t)$ can be retrieved from a Fourier transform of $\delta \mathbf{M}^S(\omega) = \chi^S(\omega) \mathbf{E}(\omega)$. The induced spin polarization follows the driving field, but it has a phase difference due to the imaginary SREE susceptibility. An equivalent relation holds for the orbital polarization. The induced spin and orbital polarizations at a frequency $\omega$ will thus still provide staggered torques on the existing static moments, but these torques will alternate with

time. A major question is then how fast the switching of the static moments can proceed, whether this can be pushed to the PHz regime. Recent experiments demonstrated that switching of CuMnAs is possible at THz frequencies[33]. Potentially, on account of the above, the switching could thus be even faster in antiferromagnets, in particular when the torques could be enhanced, but the boundaries on the switching speed are as yet unexplored.

To verify whether the SREE and/or OREE can be at the origin of ultrafast switching, and what the intrinsic frequency limit is, atomistic spin-dynamics simulations should be performed. The inclusion of both induced spin and orbital magnetic moments would notably be required to achieve the full picture. Such spin-dynamics simulations could clarify as well the role of the non-staggered, homogeneous components for the switching and the influence of Joule heating, inherently present in all experiments. It was shown recently that Joule heating plays an essential role as it drastically decreases the required switching field and enhances the spin–orbit torque efficiency[49]. Also for $Mn_2Au$, it was lately concluded that Joule heating can provide a sufficient thermal activation for switching processes[31]. It should be emphasized, too, that the switching dynamics of an antiferromagnet is distinct from that of a ferromagnet, since the magnetization dynamics of an antiferromagnet is described by a second-order differential equation, which contains a magnetic inertia term for the spins[50–52]. Antiferromagnetic inertia can provide an important stimulus for the switching, because, even after the pulse is switched off, the already induced torques will act for a longer time as drivers of the dynamics.

Switching in antiferromagnets is believed to be due to locally staggered spin–orbit fields that drive opposite dynamics of moments on the two AFM sublattices[28–34,36]. Our investigation strongly supports that the REE is an excellent candidate to explain the microscopic origin of such staggered fields. Beyond this, we report several surprising discoveries: first, there exists a significant OREE that can be much larger than the SREE. Second, we find that there exists not only staggered but also non-staggered components to the REE tensors. In both CuMnAs and $Mn_2Au$, we find that the staggered response is strongest. This causes a locking of the orbital momentum perpendicular to the applied field.

Computing the symmetry of the induced polarizations with respect to an in-plane electric field, we find that these can have a Rashba-like or a Dresselhaus-like texture and that these textures can in general be distinct for the induced spin and orbital polarizations; for example, a Dresselhaus-like symmetry for the SREE and a Rashba symmetry for the OREE of $Mn_2Au$ with in-plane AFM moments. As a consequence, the spin and orbital fields can enhance each other or cancel each other, i.e., act in a cooperative or a non-cooperative way for switching of the sublattice magnetizations.

The most surprising part of this work is undoubtedly the strong induced orbital polarization, which can be much larger than the induced spin dipole magnetization. The nonequilibrium orbital magnetization is notably even present in the absence of spin–orbit interaction. This implies that it does not arise from a small relativistic effect, but has a more fundamental, nonrelativistic origin, allowed by the $\mathcal{PT}$ symmetry of the two Mn inversion partners. While our focus here has been on the two antiferromagnets that are of current interest for antiferromagnetic spintronics, the large dominant orbital fields could gain importance in the emerging field of spinorbitronics[53]. As the induced spin and orbital polarizations originate from the coupling of the electric field to the electron charges, these induced polarizations can moreover be driven at high frequencies, opening prospects for achieving spintronics at petahertz frequencies.

Lastly, on a more general note, the here-developed general ab initio framework can be employed for the study of nonequilibrium electric-field induced polarizations in a wide range of materials, as e.g., bulk compounds and metal/ferromagnet or metal/antiferromagnet interfaces. While bulk materials can already display rich spin–orbit-related physics, interfaces of a heavy metal with a magnetic layer, where the SOC is increased at the interface, can feature an enhanced REE that can e.g., be utilized to control the spin orientation in the magnetic layer[40]. Our ab initio theory framework can provide a materials' specific understanding of the mechanisms behind electrical spin control and lead to the design of suitable interfaces for future spintronics applications.

## Methods

**Linear-response-theory formulation**. We employ linear-response theory to derive the magnetic polarization induced by a time-varying potential $\hat{V}(t) = -e\,\mathbf{E}(t)\cdot\hat{\mathbf{r}}$ that is treated as a perturbation to the time-independent Kohn–Sham Hamiltonian $\hat{H}_0$ for a periodic crystal potential. The Rashba-Edelstein tensors can be expressed in terms of the solutions of the unperturbed Kohn–Sham Hamiltonian as

$$\chi_{ij}^B(\omega) = \frac{ie}{m_e}\int_\Omega \frac{d\mathbf{k}}{\Omega}\sum_{n\neq m}\frac{f_{m\mathbf{k}}-f_{n\mathbf{k}}}{\hbar\omega_{nm\mathbf{k}}}\frac{B_{nm\mathbf{k}}^i\,p_{nm\mathbf{k}}^j}{\omega-\omega_{nm\mathbf{k}}+i\tau_{inter}^{-1}} - \frac{ie}{m_e}\int_\Omega \frac{d\mathbf{k}}{\Omega}\sum_n \frac{\partial f_{n\mathbf{k}}}{\partial\epsilon}\frac{B_{nn\mathbf{k}}^i\,p_{nn\mathbf{k}}^j}{\omega+i\tau_{intra}^{-1}}, \quad (3)$$

where $f_{n\mathbf{k}}$ is the occupation of Kohn–Sham state $|n\mathbf{k}\rangle$ with energy $\epsilon_{n\mathbf{k}}$ at wavevector $\mathbf{k}$, $\Omega$ the Brillouin zone volume, $p_{nm\mathbf{k}}^j$ are the momentum-operator matrix elements and $B_{nm\mathbf{k}}^i$ are the matrix elements of the spin ($\hat{\mathbf{S}}$) or orbital angular momentum ($\hat{\mathbf{L}}$) operator, respectively, and $\hbar\omega_{nm\mathbf{k}} = \epsilon_{n\mathbf{k}} - \epsilon_{m\mathbf{k}}$.

The REE tensor (3) contains two distinct contributions, interband ($n\neq m$) and intraband ($n = m$) contributions. The former describe transitions between the valence and conductions states, the latter describes the response of electrons around the Fermi energy. $\tau^{-1}$ is a broadening parameter that accounts for the finite electron-state lifetime; it can be different for intraband and interband transitions. Here, for sake of simplicity, we use the same value of $\tau$ for the interband and intraband contributions (see below), because our aim is to understand the role and symmetry of the spin and OREE and not their precise value. We further note that our interband formulation is different from another recent investigation[36,37], in which the expression $\frac{f_{m\mathbf{k}}-f_{n\mathbf{k}}}{\omega_{nm\mathbf{k}}+i/\tau}$ is used, but the analytic dependence used in Eq. (3) guarantees that the response is causal and that Kramers–Kronig relations between real and imaginary part are fulfilled, which is necessary for the REE at nonzero frequencies.

**Details on the numerical calculations**. To evaluate the frequency-dependent SREE and OREE tensors, we adopt the DFT formalism as implemented in the full-potential linearized augmented plane-wave (FLAPW) all-electron code WIEN2k[39]. The momentum matrix elements are computed with the WIEN2k package[54] while we use our own implementation for the spin and orbital-momentum matrix elements. In all of our calculations, we use the Perdew–Burke–Ernzerhof exchange correlation potential[55]. The broadening $\hbar\tau^{-1}$ is set to 0.41 eV (0.25 eV) for both intra- and interband transitions of CuMnAs ($Mn_2Au$), which was found to give realistic results for the (spin–orbit related) magneto-optical properties of metallic systems[56]. The spin and orbital responses to the electric field are computed over a whole range of frequency, i.e., we do not restrict our formalism to static electric fields ($\omega = 0$). Both the real and imaginary parts of the SREE and OREE susceptibility tensors are computed. Our relativistic DFT calculations include spin–orbit interaction consistent with Dirac theory, without resorting to any specific form of SOC, such as Rashba or Dresselhaus.

For CuMnAs, the product between the smallest muffin-tin radius $R_{MT}$ and the largest reciprocal vector $K_{max}$ was $R_{MT}\times K_{max} = 8.5$. The self-consistent spin-polarized density was computed using a $25\times 25\times 15$ Monkhorst-Pack grid[57]. The REE tensors were then computed using a $54\times 54\times 33$ k-point grid. For $Mn_2Au$, the product between the smallest muffin-tin radius $R_{MT}$ and the largest reciprocal vector $K_{max}$ was $R_{MT}\times K_{max} = 8.5$. The self-consistent spin-polarized density was computed using a $29\times 29\times 11$ Monkhorst-Pack grid[57]. The REE tensors were computed on a $80\times 80\times 31$ k-point grid. The numerical convergence of the REE tensor elements was checked by using much denser k-point grids. We adopted the experimental lattice constants in our calculations[9,26].

To access the influence of unit-cell doubling on the numerical results, we have computed the SREE and OREE tensors for a double-unit cell of antiferromagnetic CuMnAs with moments along the c-axis. The results, given in Supplementary Fig. 9a–c, demonstrate full numerical convergence and hence the stability of the computed tensors with respect to unit-cell size.

## Data availability

URL link to the depository of all relevant data is available upon reasonable request to the corresponding author.

## Code availability
The code that was used for linear-response DFT calculations is available from L.S. upon reasonable request.

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

## Acknowledgements

We thank Prof. E.I. Rashba for valuable comments. This work has been supported by the Swedish Research Council (VR), the K. and A. Wallenberg Foundation (grant No. 2015.0060), the European Union's Horizon2020 Research and Innovation Programme under grant agreement No. 737709 (FEMTOTERABYTE), and the Swedish National Infrastructure for Computing (SNIC). The calculations were performed at the PDC Center for High Performance Computing and the Uppsala Multidisciplinary Center for Advanced Computational Science (UPPMAX). Open access funding provided by Uppsala University.

## Author contributions

L.S. performed the ab initio calculations and programming with assistance of M.B. All authors discussed the results and analyzed them. L.S. and P.M.O. wrote the paper with

comments from M.B. and A.K.N. P.M.O. was responsible for the overall research direction.

## Competing interests

The authors declare no competing interests.
