## [Peer Review File · Nature Communications]

Reviewers' comments:

Reviewer #1 (Remarks to the Author):

Report on the manuscript

"Orbitally Dominated Rashba-Edelstein Effect in Noncentrosymmetric Antiferromagnets"

by L. Salemi, M. Berritta, A.K. Nandy, and P.M. Oppeneer.

While the authors address an interesting topic and generally present their results well, there remain issues with the validity of the conclusions:

1. The authors argue that different magnetic configurations (x or z axis) result in different response to the electric field, particularly with respect to the spin texture (x: Dresselhaus type; z: Rashba type). This requires explanation in terms of the band structures of the different magnetic configurations. The authors have to discuss the energy window in that the bands show spin textures.
2. The authors argue that the orbital polarization responds to the electric field, but do not analyze which orbitals play the main role. A projection of the band structure onto different atoms and orbitals will make it possible to understand the mechanism.
3. As the supercell size is expected to be critical for the spin texture and orbital polarization, it is essential to analyze how these quantities evolve with the supercell size (i.e., with the band folding).

Reviewer #2 (Remarks to the Author):

This manuscript titled "Orbitally Dominated Rashba-Edelstein Effect in Noncentrosymmetric Antiferromagnets" discusses the authors' finding of an unexpectedly large orbital contribution to the

REE susceptibility in CuMnAs and Mn₂Au, as well as the possible consequences of this discovery. Overall, the paper is well-written and well-organized.

The major claims of this paper are that CuMnAs and Mn₂As have orbital contributions to their REE tensors that dominated by an order of magnitude over the spin contributions. These materials have recently been suggested by many authors as candidates for antiferromagnetic memories. The authors cite an appropriate cadre of papers stretching back to Wadley et. al's 2016 paper in Science, tracing this technological enterprise that has emerged as one of the core threads of antiferromagnetic spintronics research in the past few years.

Since electrical switching is the ideal case of these memory devices, understanding the mechanism that couples electric field to the Neel order dynamics is crucial. The present paper does not investigate the Neel order dynamics per se, but lays down the crucial foundational element of such a research program: if we are to generate Neel order dynamics, where and how is the necessary angular momentum generated?

In that light, I believe this paper will have considerable influence on thinking in the field. First, because this paper offers a complete picture of where the REE comes from in these materials, it also offers guidelines for what to look for in other materials that might, for whatever reason, end up being superior technological substrates. Second, it offers a template for any such future investigations that may arise. (On that note: I do believe this paper gives enough detail for the reader to replicate their results.) Finally, and most concretely, it reveals the low-level complexity of engineering efficient switching mechanisms, while offering a trove of useful data for the materials in question.

I have little to criticize about this paper; it is well written, supported by solid data, and both relevant and timely to the field. I highly recommend its publication in Nature Communications.

We would like to thank the Reviewers for their careful reading of our manuscript and for their valuable comments, which we feel do contribute to improve the manuscript. Below we provide our answers to the comments of the Reviewers and list the changes that we have made to address the comments. All changes to the manuscript are highlighted in blue color. We have made some additional changes, related to language issues, as well as comments from discussions with other scientists. In addition there is now the new Supplementary Information that is mentioned at several places throughout the manuscript. A further change is related to the fact that we have used a higher $R \times K$ cut-off in the calculations which led to some changes in the spectral shapes of the tensors. We emphasize that these do however not affect the physics or the conclusions drawn in the manuscript.

Reviewer #1 (Remarks to the Author):

Report on the manuscript

"Orbitally Dominated Rashba-Edelstein Effect in Noncentrosymmetric Antiferromagnets"

by L. Salemi, M. Berritta, A.K. Nandy, and P.M. Oppeneer.

While the authors address an interesting topic and generally present their results well, there remain issues with the validity of the conclusions:

1. The authors argue that different magnetic configurations (x or z axis) result in different response to the electric field, particularly with respect to the spin texture (x: Dresselhaus type; z: Rashba type). This requires explanation in terms of the band structures of the different magnetic configurations. The authors have to discuss the energy window in that the bands show spin textures.

Answer: we agree with the Reviewer that is desirable to obtain more insight in the origin of the spin and orbital textures. It is however quite difficult to recognize these from the energy band structures. To offer more insight, we have added an explanation of the dominant orbital Rashba-Edelstein effect in terms of PT symmetry. This analysis on page 7 shows, that even without spin-orbit interaction, PT symmetry will enforce staggered orbital polarizations on the Mn atoms. The dominant Rashba-type orbital texture in these materials is a result of this PT symmetry, consistent with the calculated result in Fig. 5.

We have furthermore analyzed the size of the induced spin and orbital textures as a function of the energy window by changing the chemical potential. This shows that in general the spin and orbital polarizations are largest around the energy where there is a maximum of the Mn 3d DOS. This we have added to a new paragraph on orbital decomposition and bandfilling effects, on page 8, and a figure in the Supplementary Information. The spin and orbital textures do depend on the bandfilling and could even be tuned by changing the bandfilling.

2. The authors argue that the orbital polarization responds to the electric field, but do not analyze which orbitals play the main role. A projection of the band structure onto

different atoms and orbitals will make it possible to understand the mechanism.

Answer: this is indeed a good point of the Reviewer. We have modified the code and performed calculations to address this point. Specifically, we have performed a projection of the spin and orbital Rashba-Edelstein effects both on the constituting atoms and their orbitals. In this way we can analyze which orbitals play the main role. The projection on the orbitals shows that the d-orbitals of Mn are responsible for nearly all of the spin and orbital effects. This information can indeed be valuable for the readers and we have therefore included it in the new Supplementary Information. It does unfortunately not work well to see this information from the energy eigenvalues, i.e., the band structures. We have added a discussion of the orbital projections on page 8 and we also discuss there the effect of filling the band structures on the spin textures (Rashba versus Dresselhaus).

3. As the supercell size is expected to be critical for the spin texture and orbital polarization, it is essential to analyze how these quantities evolve with the supercell size (i.e., with the band folding).

Answer: the Reviewer is right that supercell effects could in principle affect the spin and/or orbital texture. We have computed the influence of supercell doubling on the spin and orbital Rashba-Edelstein effects and have included several of these calculations in the Supplementary Information. These calculations illustrate that the calculated Rashba-Edelstein tensors do not change with increased supercell size. A paragraph discussing this has furthermore been added to the Methods section.

Reviewer #2 (Remarks to the Author):

This manuscript titled "Orbitally Dominated Rashba-Edelstein Effect in Noncentrosymmetric Antiferromagnets" discusses the authors' finding of an unexpectedly large orbital contribution to the REE susceptibility in CuMnAs and Mn₂Au, as well as the possible consequences of this discovery. Overall, the paper is well-written and well-organized.

The major claims of this paper are that CuMnAs and Mn₂As have orbital contributions to their REE tensors that dominated by an order of magnitude over the spin contributions. These materials have recently been suggested by many authors as candidates for antiferromagnetic memories. The authors cite an appropriate cadre of papers stretching back to Wadley et. al's 2016 paper in Science, tracing this technological enterprise that has emerged as one of the core threads of antiferromagnetic spintronics research in the past few years.

Since electrical switching is the ideal case of these memory devices, understanding the mechanism that couples electric field to the Neel order dynamics is crucial. The present paper does not investigate the Neel order dynamics per se, but lays down the crucial foundational element of such a research program: if we are to generate Neel order dynamics, where and how is the necessary angular momentum generated?

Answer: yes, we completely agree with the Reviewer on this.

In that light, I believe this paper will have considerable influence on thinking in the

field. First, because this paper offers a complete picture of where the REE comes from in these materials, it also offers guidelines for what to look for in other materials that might, for whatever reason, end up being superior technological substrates. Second, it offers a template for any such future investigations that may arise. (On that note: I do believe this paper gives enough detail for the reader to replicate their results.) Finally, and most concretely, it reveals the low-level complexity of engineering efficient switching mechanisms, while offering a trove of useful data for the materials in question.

I have little to criticize about this paper; it is well written, supported by solid data, and both relevant and timely to the field. I highly recommend its publication in Nature Communications.

Answer: we are grateful to the Reviewer for these positive comments on our manuscript!

REVIEWERS' COMMENTS:

Reviewer #1 (Remarks to the Author):

The authors have answered all my questions. The manuscript is ready for publication.

We would like to thank the Reviewer for his/her careful reading of our manuscript. Our response to the recommendation of the Reviewer is given below.

REVIEWERS' COMMENTS:

Reviewer #1 (Remarks to the Author):

The authors have answered all my questions. The manuscript is ready for publication.

Answer: we are grateful to the Reviewer for this positive recommendation.